# Counselling experiences among men having sex with men and living with HIV in Malaysia

**Tuan Norbalkish Tuan Abdullah**[1], **Ruhani Mat Min**[2]*, **Siti Salina Abdullah**[2],
**Mosharaf Hossain**[2]

1 Faculty of Psychology and Education, Universiti Malaysia Sabah, Kota Kinabalu, Malaysia, 2 Faculty of Business, Economics and Social Development, Universiti Malaysia Terengganu, Kuala Terengganu, Malaysia

☯ These authors contributed equally to this work.
* ruhani@umt.edu.my

**Data Availability Statement:** Data cannot be shared publicly because the data contain potentially identifying information about the research participants. Data are available from the Medical Research and Ethics Committee of the Ministry of

## Abstract

### Purpose

In Malaysia, the trend of HIV transmission has shifted from intravenous drug use to sexual intercourse, and men who have sex with men (MSM) have become the main driver due to high-risk sexual behaviour. Thus, treatment and care, which also involves counselling, for men who have sex with men and who are living with HIV (MSM living with HIV) are crucial. This study aims to explore the experiences of MSM living with HIV and participating in counselling session during treatment and care at two public hospitals.

### Method

This qualitative study with a grounded-theory approach was conducted at two public hospitals in Malaysia. Five participants who were MSM living with HIV were selected through purposive sampling. They participated in semi-structured interviews, non-participant observations, and diary entries, each of which was conducted three times. The data were analysed using grounded theory with N-Vivo 8 to determine themes.

### Result

The participants were found to experience feelings of emptiness and hopelessness because of their unreadiness to accept their HIV status. These feelings made their participation in counselling sessions challenging. Consequently, the participants found counselling sessions unhelpful due to their unwillingness to participate in the counselling relationship.

### Conclusion

The findings of the study highlight the need for counselling sessions to focus more on feelings related to unreadiness to improve the self-esteem and ability to create positive relationships with others of MSM living with HIV. It is also important to strengthen the training and skills among HIV counsellors to enhance counselling services for these men.

Health Malaysia (contact via mrecsec@moh.gov.my or MREC officer in-charge: Dr. Christie A/P Machial (drchristie@moh.gov.my) for researchers who meet the criteria for access to confidential data.

**Funding:** The authors received no specific funding for this work.

**Competing interests:** The authors have declared that no competing interest exist.

## Introduction

In recent years, men who have sex with men and who are living with HIV (MSM living with HIV) have become the major concern in the HIV epidemic, as global reports have revealed a high prevalence of HIV among this key population [1]. New diagnoses of HIV among this group are reported to have increased to more than 50% in most European countries, and to about 25% in Asia and the Pacific, since 2017 [1]. Globally, MSM are identified as 27 times more likely to acquire HIV than the general population because of high-risk sexual behaviour [1, 2]. In Malaysia, the trend of HIV transmission has shifted from intravenous drug use to sexual intercourse since 2011, and MSM have been reported as the main driver of the epidemic since 2018 [2]. Consequently, HIV prevention efforts in Malaysia have become critical.

Moreover, it has been reported that in 2017, less than 50% of people living with HIV (PLHIV) participated in HIV treatment, and adherence to HIV prevention programmes among MSM living with HIV was only 37.4% [2]. Factors contributing to the vulnerability of MSM living with HIV are related to stigma and hidden identity [1–3]. Aspects of stigma include homophobic stigma, discrimination, violence, and fear of negative reactions from healthcare workers, all of which drive members of this population to hide their identities and sexual orientations because of their unreadiness to accept their HIV status [1]. In addition, fear, including the fear of negative reactions, affects their self-esteem [4]. Thus, these challenges discourage members of this group from accessing treatment programmes because of the fear of their identities being revealed [1, 5]. Previous studies have found that stigma contributes to MSM maintaining hidden identities and experiencing career challenges, and prevents them from undergoing treatment programmes concerning HIV prevention [5–8]. Therefore, these factors are also seen as barriers for MSM living with HIV to engage with support services provided by HIV prevention programmes.

MSM are widely exposed to behavioural risks of HIV infection because they have multiple sexual partners and have been found to consistently engage in casual sex without using condoms [4, 9]. Previous studies have shown that the main cause of high-risk behaviour among MSM living with HIV is their lack of knowledge of HIV and awareness of protection during sexual activities [10, 11]. Hence, participation in HIV treatment and care programmes is extremely important for members of this HIV group, as it would improve their knowledge and awareness of the virus. Counselling services are a part of HIV treatment and care for those living with HIV [12–14], and may similarly help improve their knowledge and awareness. However, only a limited number of MSM living with HIV are involved in treatment and care prevention programmes [2]. This is related to their perceptions of themselves: MSM living with HIV and experiencing stigma might perceive themselves as failures [1, 4, 5], a perception that is related to their inability to fulfil their basic needs [4]. In addition, MSM living with HIV may experience unsatisfying relationships [4], as counsellors have reported limited relationship knowledge and related skills among PLHIV [15].

Given these issues, MSM living with HIV have limited experience with counselling services. Therefore, this study aimed to explore and understand the experiences of MSM living with HIV in engaging with counselling services in the hospital setting.

## Method

### Study design, ethics statement, and participants

This qualitative study with a grounded-theory approach was conducted at two public hospitals in Malaysia between March 2017 and February 2018. These hospitals were selected because they had certain numbers of PLHIV who were reported to regularly attend and engage in HIV

treatment and counselling services. The two hospitals were selected as locations for this research because they were expected to yield different experiences within and across the sites. Ethical approval to conduct the study was obtained from the Medical Research and Ethics Committee of the Ministry of Health of Malaysia.

The researchers gathered data via semi-structured interviews, non-participant observation, and diary entries. Using purposive sampling, five participants who were MSM living with HIV were selected based on inclusion and exclusion criteria. Qualifying participants were over 18 years old, literate, and had been diagnosed with HIV at least two years before the study commenced. The participants also regularly attended counselling sessions, which are part of the HIV treatment procedures at the hospitals.

The researcher (TN) conducted an initial face-to-face meeting with each potential participant, and not as part of the treatment, to explain the purpose, ethical principles, and the duration of the study to obtain voluntary participation. During this session, the participants disclosed their HIV status to TN, who was not affiliated with the hospitals. Awareness of the confidentiality required was demonstrated throughout this research [16, 17]. Written informed consent was obtained from the participants, who had not experienced threats, injustice, or manipulation [16–18]. The participants' backgrounds are summarised in Table 1.

## Data collection

From March 2017 to February 2018, TN, a registered female counsellor, conducted semi-structured interviews, non-participant observations, and diary-writing sessions three times for each participant. The researchers used multiple methods in data collection, as this was expected provide a better understanding of the phenomena being studied and increase the trustworthiness of the data [16–18]. Five MSM living with HIV participated in the data gathering process, which ended when the category of experiences in engaging with the counselling session was saturated [16, 19].

**Table 1. Participants' backgrounds.**

| Background information | Number | Percentage (%) |
|---|---|---|
| **Age (years)** | | |
| 30–40 | 3 | 60 |
| >40 | 2 | 40 |
| **Education** | | |
| Bachelor's degree | 3 | 60 |
| Master's degree | 2 | 40 |
| **Occupation** | | |
| Employed | 4 | 80 |
| Businessman | 1 | 20 |
| **Income (MYR)** | | |
| 2,000–4,000 | 3 | 60 |
| >4,000 | 2 | 40 |
| **Duration of HIV treatment (years)** | | |
| <5 | 3 | 60 |
| 5–10 | 2 | 40 |
| **Duration of attending counselling sessions (years)** | | |
| <5 | 3 | 60 |
| 5–10 | 2 | 40 |

In this study, semi-structured interviews provided the main data, which were obtained through one-to-one interactions between the researcher and the participants based on a set of open-ended questions formulated in advance and in variable sequence [20, 21]. This method was intended to ensure that all the issues were covered in adequate depth during the interview sessions [16, 17, 21]. Details of the interview questions appear in Table 2.

The participants were made aware of the purpose of the study before the interview sessions. The participants were allowed to decide the time and date for the interviews to ensure freedom from elements of threat [19–21]. The interviews were conducted using Bahasa Malaysia, which was the spoken language of the participants [17], and they were audio recorded with the consent of the participants. During the interviews, the participants were given time to respond to the open-ended questions, which allowed them to share their experiences using their own words [17, 21].

The researchers also collected data using the observational method, in which non-participant observations were conducted to describe the setting, activities, and participants of the study [16, 17]. The non-participant observations were carried out to observe the counselling sessions attended by the participants at the two selected public hospitals. During the observations, a researcher sat in the same counselling room, but the presence of the researcher was not noticeable to the participants because the researcher was separated from the counselling area by a bookshelf. The researcher was aware of issues of confidentiality during the procedure.

The participants were also required to complete a diary by responding to prepared and open-ended statements (Table 3); they were free to select their own words and style of writing [17, 20]. By writing about their experiences in engaging with counselling services at hospitals, they created private documents that represented their thoughts, feelings, opinions and actions [20]. This was also done to enable the researchers to obtain participants' own language and words [16, 19, 20].

Table 2. Interview questions.

| Interview number | Questions |
| --- | --- |
| 1 | 1. How would you describe your background? |
| | 2. How would you describe your current situation? |
| | 3. What kind of feelings have you experienced since you became aware that you are HIV positive? |
| | 4. Tell me about your life after you discovered you are HIV positive. |
| | 5. How did you define yourself after you discovered you are HIV positive? |
| 2 | 1. What changes have you faced since you discovered you are HIV positive? |
| | 2. Tell me about your experiences in facing the changes after you discovered you are HIV positive. |
| | 3. What are your feelings about facing the changes? |
| | 4. What can you tell me about the challenges you have faced since you discovered you are HIV positive? |
| | 5. What are your feelings about facing the challenges after you discovered you are HIV positive? |
| | 6. How do you see yourself facing the challenges? |
| 3 | 1. Tell me about your expectations about your life after being infected with HIV. |
| | 2. What were your ideas about your life since you discovered you are HIV positive? |
| | 3. What activities have you engaged in since you discovered you are HIV positive? |
| | 4. What are your interests in life? |
| | 5. What were your feelings after having been through the treatment and counselling session? |
| | 6. What are your strengths? |

**Table 3. Prepared and open-ended statements for diary writing.**

| No | Items |
|---|---|
| 1 | After engaging in the counselling session, my feelings are. . . |
| 2 | During the counselling session, I am doing. . . |
| 3 | My feelings during the activities. . . |
| 4 | What I learn from this counselling session. . . |
| 5 | My view of counselling services. . . |

## Data analysis

The grounded-theory approach was applied to analyse the data, from which categories, themes, and patterns emerged [21, 22]. The process of analysis was performed by TN, and begun immediately after the initial interviews. It followed each interview, observation, and diary-writing activity thereafter. The interviews and diary entries were transcribed into English, and N-VIVO 8 was used to identify the themes related to the participants' experiences. Triangulation of the three main forms of data and the various aspects of participants' experiences increased the trustworthiness and credibility of the findings, which allowed for alternative interpretations [16, 22–24].

## Results

The responses of the participants, MSM living with HIV, indicated that they were uncomfortable and demotivated, which disconnected them from relationships with others and led to a lack of interest in their lives, thus causing feelings of emptiness. The findings also indicated a lack of hope and a lack of trust, which discouraged the participants from leading meaningful lives and led to feelings of hopelessness. All five participants reported these experiences (Table 4).

As mentioned earlier, counselling is a part of treatment and care programmes for PLHIV at the selected hospitals. In other words, the participants were required to attend counselling sessions during the treatment and care programmes. The participants brought those feelings of emptiness and hopelessness to the counselling sessions, which affected their experiences of counselling.

### Feelings of emptiness

The participants expressed feelings discomfort and demotivation, and these two feelings are central to the feeling of emptiness. The expectation of rejection, concealment, and internalised homophobia were the factors underlying feelings of discomfort and demotivation [25, 26]. In this study, the participants expressed feeling uncomfortable with discussing their experiences. They did not want to reveal their true selves due to their expectations of rejection. For example,

**Table 4. Counselling experience among MSM living with HIV.**

| Participant | Feeling of emptiness | | Feeling of hopelessness | |
|---|---|---|---|---|
| | Uncomfortable | Demotivated | No hope | No trust |
| Participant 1 | / | / | / | / |
| Participant 2 | / | / | / | / |
| Participant 3 | / | / | / | / |
| Participant 4 | / | / | / | / |
| Participant 5 | / | / | / | / |

Participant 1 felt uncomfortable sharing his stories in the counselling session, and he felt doing so could not change his status. In other words, he felt uncomfortable and demotivated due to his status as a man who has sex with men and is living with HIV:

> I was asked to attend the session every time I attended the treatment. Honestly, I was not comfortable. That does not mean counselling is not good, but the feeling remains after I left the counselling session. For me, it was nothing and useless because I know there will be no change with HIV. (Participant 1)

Participants 2 and 3 also shared feelings of discomfort and demotivation due to their inability to accept their HIV status:

> I don't think that would be easy for me to adapt with a service like counselling, even though I know they just want to help me. But I am blank with this disease. I had never expected HIV in my body now and my life will end with this. (Participant 2)

> I understood that counselling is to help and to motivate people, but this is HIV, and I still cannot understand why they ask me to attend the counselling session. It is nothing for me. There is no one, and nothing could change this. It is HIV and I know that my life will end soon. (Participant 3)

During the non-participant observation, Participants 1 and 3 were observed to behave passively during the counselling sessions, as indicated by their poor response to the counsellor.

During the diary writing, Participant 2 wrote in his entries that he was aware of his life becoming meaningless with HIV: 'There is nothing more in my life. I am very sure that I will end up with everything in a terrible condition.' Participant 4 wrote in his diary that being a man who has sex with men and who is living with HIV obliterated his happiness in life, which explained his feeling of emptiness:

> HIV has completely taken over everything in my life. I have lost my happiness. My family, friends and career. I still have a job, but I know it will end at any time with the deteriorating health, painful medication and struggles that I need to face outside. I have attended counselling sessions, but there is no hope for me. I am totally broken. (Participant 4)

On the other hand, Participant 5 shared that he attended the counselling session just to discuss the treatment and medicines. Participating in counselling session is a requirement of the treatment and care programmes. Although he participated in counselling sessions, he still felt demotivated, which prevented counselling from having a positive effect and providing support to him as a man who has sex with men and who is living with HIV:

> I only felt nothing in the counselling session. It does not mean that I do not like the counsellor, but I think that they could not help much. Yes, I asked and discussed a lot of things related to medicines and treatments, but the feeling that I have inside remained same. I am always thinking of death and there are still many things I want to grab in my life. (Participant 5)

The participants reported feeling uncomfortable and demotivated while attending the counselling sessions at hospitals. Because of those feelings, they chose to not share their stories with the counsellors. They might have acted in a similar way with other people, and their withdrawing from others may have led to a lack of relationships, resulting in a feeling of emptiness.

This feeling may have also been related to their inability to accept their HIV status and their frustration with their diagnosis, which prevented them from benefitting from counselling.

## Feelings of hopelessness

Hopelessness among MSM living with HIV is an aspect of psychological distress that contributes to non-adherence to medication regimes and poor engagement with society [27, 28]. In this study, all the participants were found to have no hope in life after being diagnosed with HIV. They also reported having no trust in those who wanted to help them. Lacking hope and trust is central to hopelessness. Participant 1 did not trust his counsellor, and was thus unwilling to discuss his life as a man who has sex with men and is living with HIV and ways to handle his emotions related to the experience:

> I admit that I am not the type of person who easily shares anything about myself. In the counselling sessions, I noticed and could feel that the counsellor tried to get some information about me, for example, my personal issues. For me, it is not easy to share and to make people understand it. My situation will never get better. No one can help me. So, I only chose to discuss treatments. (Participant 1)

During the non-participant observations, Participant 1 was observed to be uncomfortable talking during the counselling session. The counsellor hardly addressed his discomfort, and he was seen as reluctant to respond.

Participant 4 expressed similar feelings of having no hope:

> HIV has changed my life. I only feel like giving up on everything. It is too late now. I have no future living with this disease. I started to have no mood for working and doing the things I used to do. I will never come back to the way I was. (Participant 4)

In addition, Participant 4 was observed ignoring the counsellor when he was asked about his personal issues related to family and his daily activities.

Participant 3 wrote in his diary that he would be more comfortable sharing personal stories with people close to him, and not sharing anything in a professional session such as counselling:

> Honestly, it is not easy for me to continue life with HIV, and I believe there is no one who will accept and understand me now. With HIV, I need more space for myself. I prefer to share with the ones that I am close to, but not in the counselling session. I know that the counsellor tried to help me, but in terms of dealing with personal matters, I could not make it in counselling session. Sometimes, I'm more comfortable talking with a medical doctor. (Participant 3)

The participants reported feeling no hope and no trust, which are the central aspects of hopelessness. Because of those feeling, they kept their dissatisfaction about their situation, as MSM living with HIV, to themselves. This may have deterred them from obtaining adequate support from treatment and care programmes for PLHIV.

## Discussion

The findings revealed that the participants experienced feelings of emptiness and hopelessness in the counselling sessions, which explained their unwillingness to share their experiences as

MSM living with HIV. Those feelings hindered them from building counselling relationships with their counsellors, which affected their engagement in the counselling sessions [4]. Participants 1, 3 and 4 stated they felt uncomfortable when they attended the counselling sessions because they did not want the counsellors to know their experiences as MSM living with HIV. They perceived that no one would accept them and that there was no way to change their status. It is possible that those feelings of emptiness and hopelessness were related to their unreadiness to accept their reality as MSM living with HIV. In addition, they may have perceived themselves as failures [4] due to their situations as MSM living with HIV.

The feelings of emptiness and hopelessness also led to low self-esteem. This study supported the previous findings that low self-esteem among MSM living with HIV contribute to depression because of low motivation [25, 26]. This finding adds another dimension of understanding of the source of depression among MSM living with HIV, which is related to discomfort with discussing their situation, feeling demotivated, and a lack of hope and trust. Another key explanation of low self-esteem is that the respondents were not ready to accept themselves as MSM living with HIV. This further supported previous studies that found that depression levels among MSM living with HIV affect their engagement with and adherence to HIV prevention programmes because of their failure to build trusting relationships with others [1, 10, 27–29].

The participants attended the counselling sessions while feeling empty and hopeless, as they were aware of and understood the effects of HIV on their lives. Therefore, they needed to be helped and supported in dealing with those feelings. As mentioned by the participants, their lives as MSM living with HIV could not be changed. However, effective support could help them to manage their negative feelings, such as demotivation and a lack of trust in others, and accept their lives in positive ways. As MSM living with HIV, their realities will be different from those of others. They need to accept their condition and then create their own perspectives on fulfilling their basic needs in life [4]. The participants shared that they were aware of the roles of the counsellors in helping them manage their lives with HIV, but they felt uncomfortable sharing much about their lives. Those feelings deterred them from building a trusting relationship with their counsellors, and may have caused their failure to adhere to the HIV treatment and care programmes. Their inability to form counselling relationships may also contribute to their failure to create meaningful connections with others [4, 30], which may later affect their daily lives as PLHIV. This is also supported by studies that found that MSM living with HIV are a passive group in HIV prevention programmes who are affected by depression and social disconnection [1, 10, 28], which negatively affect daily life.

This study highlights the need for counselling sessions to focus more on feelings related to unreadiness to accept HIV status among MSM living with HIV. Those feelings hindered the participants from building counselling relationships with the counsellors, which affected their engagement in the counselling sessions and adherence to HIV prevention programmes. The limitation of the study is that the findings cannot be generalised to broader populations of MSM living with HIV with the same degree of certainty, because those findings have not been tested to determine whether they were statistically important or due to chance.

## Conclusion

This study demonstrates how feelings of emptiness and hopelessness contribute to MSM living with HIV viewing counselling as unhelpful. These feelings might relate to their unreadiness to accept their status, which might affect their adherence to HIV treatment and care programmes. Thus, the study highlights the need for counselling services to focus on feelings related to this unreadiness among MSM living with HIV. Their acceptance of their HIV status

is extremely important to improve their commitment to HIV treatment and care programmes. It is also crucial to strengthen the training and skills of HIV counsellors to enhance the effects of counselling among MSM living with HIV.

## Supporting information

**S1 Data.**
(DOCX)

**S1 File.**
(DOCX)

## Acknowledgments

We would like to thank the facility leaders and providers, who provided immense support to our teams undertaking the study. We are also immensely grateful to the participants who willingly participated in the study and patiently shared their experiences with our teams.

## Author Contributions

**Data curation:** Tuan Norbalkish Tuan Abdullah.

**Formal analysis:** Tuan Norbalkish Tuan Abdullah.

**Methodology:** Ruhani Mat Min, Mosharaf Hossain.

**Supervision:** Ruhani Mat Min, Siti Salina Abdullah, Mosharaf Hossain.

**Writing – original draft:** Tuan Norbalkish Tuan Abdullah.

**Writing – review & editing:** Ruhani Mat Min, Siti Salina Abdullah, Mosharaf Hossain.

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
