## [Decision Letter · Decision Letter 0]

29 Nov 2021

PONE-D-21-08579Counselling experiences among HIV men who have sex with men in MalaysiaPLOS ONE

Dear Dr. Mat Min,

Thank you for submitting your manuscript to PLOS ONE. After careful consideration, we feel that it has merit but does not fully meet PLOS ONE’s publication criteria as it currently stands. Therefore, we invite you to submit a revised version of the manuscript that addresses the points raised during the review process.

We look forward to receiving your revised manuscript.

Kind regards,

Pande Putu Januraga, M.D., DrPH

Academic Editor

PLOS ONE

2. Please include a copy of the interview guide used in the study, in both the original language and English, as Supporting Information, or include a citation if it has been published previously.

3. Thank you for stating in the text of your manuscript "Written informed consent was obtained from the participants, and they were free from threats, injustice or manipulation". Please also add this information to your ethics statement in the online submission form.

4. In your Methods section, please provide additional information about the participant recruitment method and the demographic details of your participants. Please ensure you have provided sufficient details to replicate the analyses such as:

a) the names of the hospitals patients were from,

b) a description of any inclusion/exclusion criteria that were applied to participant recruitment,

c) a statement as to whether your sample can be considered representative of a larger population, and

d) a description of how participants were recruited.

Additional Editor Comments:

Dear Authors,

Thank you for submitting your paper to Plos One.

We have secured a review from an external expert.

Would you please read the comments provided carefully and revise the paper accordingly?

The submission of the revised paper will not guarantee acceptance; we have to send the manuscript for the second round review.

I want to suggest the authors refer to the COREQ guideline in revising the manuscript. The author may submit a filled COREQ form with the manuscript for review.

All the best,

Pande

Reviewers' comments:

Reviewer's Responses to Questions

**Comments to the Author**

1. Is the manuscript technically sound, and do the data support the conclusions?

Reviewer #1: No

2. Has the statistical analysis been performed appropriately and rigorously? 

Reviewer #1: N/A

3. Have the authors made all data underlying the findings in their manuscript fully available?

Reviewer #1: No

4. Is the manuscript presented in an intelligible fashion and written in standard English?

Reviewer #1: No

5. Review Comments to the Author

Reviewer #1: Counselling experiences among HIV men who have sex with men in Malaysia

This is a paper that attempts to describe the experiences of men who have sex with men living with HIV. Overall, the manuscript is scanty and requires details to understand the context of counseling sessions. The number of participants in this study limits the interpretation of the results and the conclusion. Had the background on the study participants been provided, then likely would have explain the small sample. A description of how the authors were able to achieve saturation in this study is not provided neither is this cited as a limitation.

The overall paper would benefit from copy-editing.

Title: The title is ok, however reference should be made to men who have sex with men living with HIV and not HIV MSM.

Abstract: Introduction section is not clear what the problem is. Is increasing HIV incidence among MSMs associate with their experiences with counseling services? Methods section needs rewording;

Introduction: The section has several bold claims that are not substantiated. Line 50 has no reference. The sentence 51-54 referencing a WHO STI fact sheet (https://www.who.int/news-room/fact-sheets/detail/sexually-transmitted-infections-(stis)) does not seem to have relevant info on MSMs neither does it provide the estimates in European countries. It is unclear the % provided are increasing from what or when?

Line 61 –Not clear what “…adherence to HIV prevention programmes among HIV MSM was less than 70% dropping to 37.4% in 2017” means

Line 66 – Reference at the end (32) either is incorrect or the whole references in the document is a miss.

Difficult to follow what the problem is and what this manuscript is contributing to the problem. In general the introduction needs to be revised. The authors should provide a background and the problem that this paper is responding to. What is the prevalence? Incidence? of HIV in Malaysia? What is the driver of the HIV incidence? What HIV counseling guidelines are available in Malaysia. Which of these is this manuscript focusing on, and why is it important?

Methods: The Methods section is mixed with the results (participants’ characteristics). The selection of the hospital is so vague – “these hospitals were selected because they have certain PLWH who were reported to regularly attend and engage….”. How were participants approached and selected to participate? Was researcher (TN) part of the counsellors providing counseling services in any of the two hospitals? Some of the interview question are so ambiguous? Could this be due to direct translation from Bahasa Malaysia to English? For example; “How would you describe your current situation?” In what context?

Results: There are claims in the results that are not supported by the extracts – line 155 “the stories of the HIV MSM……that they were not ready to accept the status of their HIV”. This claim is carried through the findings. Another claim made and not supported by the extract is in line 159-160 “This is due to the poor trust towards self and others, affecting their motivation to live as HIV MSM”. The effect on motivation to live with HIV is not substantiated by the information provided. Trust for others seemed to be there for example in line 230-234, the participant shares their preference to share information to people close to them or their medical doctor.

The theme on hopelessness is not supported by any of the extracts – most if not all denotes trust issues.

This study would have benefited from interviewing counselors to understand from their perspectives the processes of counseling and corroborate the information with those of the MSMs.

Discussion and conclusion: This section requires an overhaul should this manuscript be considered for publication. The basis of the discussion is skewed as it is not supported by the findings. The claim that the participants have feelings of emptiness and hopelessness in the counseling sessions because of their inability to accept their HIV status seems to be farfetched.

In summary, while this study has important findings that could inform how counseling should be tailored to improve care and positive living among MSMs, the manuscript falls short of presenting and arguing this case. It reads as thought the authors already had in mind what they wanted to share and were looking for data that can support their claim instead of the other way round.

6. PLOS authors have the option to publish the peer review history of their article (what does this mean?). If published, this will include your full peer review and any attached files.

Reviewer #1: No

---

## [Author Response · Author response to Decision Letter 0]

16 Feb 2022

Response to reviewers

1 Please ensure that your manuscript meets PLOS ONE's style requirements, including those for file naming. The PLOS ONE style templates can be found at 

Done

2 Please include a copy of the interview guide used in the study, in both the original language and English, as Supporting Information, or include a citation if it has been published previously. 

Attached 

3 Thank you for stating in the text of your manuscript "Written informed consent was obtained from the participants, and they were free from threats, injustice or manipulation". Please also add this information to your ethics statement in the online submission form. 

Done

4 The names of the hospitals patients were from 

The names of the hospital cannot mention in the manuscript due to confidentiality. 

These two are public hospital (fully funded by the government) and located at the peninsular of Malaysia.

5 A description of any inclusion/exclusion criteria that were applied to participant recruitment 

Done, page 5, line 73-78

6 A statement as to whether your sample can be considered representative of a larger population, and 

This sample cannot consider as representative of a larger population. 

7 A description of how participants were recruited. 

Done, page 5, line 80-82

8 Counselling experiences among HIV men who have sex with men in Malaysia

This is a paper that attempts to describe the experiences of men who have sex with men living with HIV. Overall, the manuscript is scanty and requires details to understand the context of counseling sessions. The number of participants in this study limits the interpretation of the results and the conclusion. Had the background on the study participants been provided, then likely would have explain the small sample. A description of how the authors were able to achieve saturation in this study is not provided neither is this cited as a limitation. 

Rewrite has been done and it marked in blue.

This is a qualitative study with a grounded-theory approach and saturation has been mentioned (page 6, line 95-96)

9 The overall paper would benefit from copy-editing. 

Done 

10 The title is ok, however reference should be made to men who have sex with men living with HIV and not HIV MSM. 

Correction has been done and marked in blue

11 Abstract: Introduction section is not clear what the problem is. Is increasing HIV incidence among MSMs associate with their experiences with counseling services? Methods section needs rewording; 

Correction has been done and marked in blue

12 Introduction: The section has several bold claims that are not substantiated. Line 50 has no reference. The sentence 51-54 referencing a WHO STI fact sheet (https://www.who.int/news-room/fact-sheets/detail/sexually-transmitted-infections-(stis)) does not seem to have relevant info on MSMs neither does it provide the estimates in European countries. It is unclear the % provided are increasing from what or when? 

Correction has been done and marked in blue

13 Line 61 –Not clear what “…adherence to HIV prevention programmes among HIV MSM was less than 70% dropping to 37.4% in 2017” means 

Rewrite of this statement has been done and marked in blue

14 Line 66 – Reference at the end (32) either is incorrect or the whole references in the document is a miss. 

Correction has been done and marked in blue

15 Difficult to follow what the problem is and what this manuscript is contributing to the problem. In general the introduction needs to be revised. The authors should provide a background and the problem that this paper is responding to. What is the prevalence? Incidence? of HIV in Malaysia? What is the driver of the HIV incidence? What HIV counseling guidelines are available in Malaysia. Which of these is this manuscript focusing on, and why is it important? 

Rewrite has been done and marked in blue

16 Methods: The Methods section is mixed with the results (participants’ characteristics). The selection of the hospital is so vague – “these hospitals were selected because they have certain PLWH who were reported to regularly attend and engage….”. How were participants approached and selected to participate? Was researcher (TN) part of the counsellors providing counseling services in any of the two hospitals? Some of the interview question are so ambiguous? Could this be due to direct translation from Bahasa Malaysia to English? For example; “How would you describe your current situation?” In what context? 

Rewrite has been done and marked in blue.

A copy of interview questions, in English and Bahasa Malaysia is attached.

17 Results: There are claims in the results that are not supported by the extracts – line 155 “the stories of the HIV MSM……that they were not ready to accept the status of their HIV”. This claim is carried through the findings. Another claim made and not supported by the extract is in line 159-160 “This is due to the poor trust towards self and others, affecting their motivation to live as HIV MSM”. The effect on motivation to live with HIV is not substantiated by the information provided. Trust for others seemed to be there for example in line 230-234, the participant shares their preference to share information to people close to them or their medical doctor.

The theme on hopelessness is not supported by any of the extracts – most if not all denotes trust issues. 

Rewrite has been done and marked in blue.

18 This study would have benefited from interviewing counselors to understand from their perspectives the processes of counseling and corroborate the information with those of the MSMs. 

Thank you for the suggestion. Please refer to reference no 13.

19 Discussion and conclusion: This section requires an overhaul should this manuscript be considered for publication. The basis of the discussion is skewed as it is not supported by the findings. The claim that the participants have feelings of emptiness and hopelessness in the counseling sessions because of their inability to accept their HIV status seems to be farfetched 

Rewrite has been done and marked in blue.

20 In summary, while this study has important findings that could inform how counseling should be tailored to improve care and positive living among MSMs, the manuscript falls short of presenting and arguing this case. It reads as thought the authors already had in mind what they wanted to share and were looking for data that can support their claim instead of the other way round. 

Rewrite has been done and marked in blue.

---

## [Decision Letter · Decision Letter 1]

12 Jul 2022

PONE-D-21-08579R1Counselling experiences among men sex men living with HIV in MalaysiaPLOS ONE

Dear Dr. Mat Min,

Thank you for submitting your manuscript to PLOS ONE. After careful consideration, we feel that it has merit but does not fully meet PLOS ONE’s publication criteria as it currently stands. Therefore, we invite you to submit a revised version of the manuscript that addresses the points raised during the review process. Please note that we have only been able to secure a single reviewer to assess your manuscript. We are issuing a decision on your manuscript at this point to prevent further delays in the evaluation of your manuscript. Please be aware that the editor who handles your revised manuscript might find it necessary to invite additional reviewers to assess this work once the revised manuscript is submitted. However, we will aim to proceed on the basis of this single review if possible. 

We look forward to receiving your revised manuscript.

Kind regards,

Thomas Tischer

Staff Editor

PLOS ONE

Journal Requirements:

Additional Editor Comments (if provided):Please address the reviewers comments and add a detailed section about the limitations of the study 

Reviewers' comments:

Reviewer's Responses to Questions

**Comments to the Author**

1. If the authors have adequately addressed your comments raised in a previous round of review and you feel that this manuscript is now acceptable for publication, you may indicate that here to bypass the “Comments to the Author” section, enter your conflict of interest statement in the “Confidential to Editor” section, and submit your "Accept" recommendation.

Reviewer #1: All comments have been addressed

2. Is the manuscript technically sound, and do the data support the conclusions?

Reviewer #1: Yes

3. Has the statistical analysis been performed appropriately and rigorously? 

Reviewer #1: N/A

4. Have the authors made all data underlying the findings in their manuscript fully available?

Reviewer #1: Yes

5. Is the manuscript presented in an intelligible fashion and written in standard English?

Reviewer #1: Yes

6. Review Comments to the Author

Reviewer #1: The revisions in this manuscript has improved the readability and understanding.

The discussion section should include a paragraph on the strengths and weaknesses/limitation of this study. It is alluded to in the last sentence of the conclusion, although, I feel this is placed in the wrong section.

7. PLOS authors have the option to publish the peer review history of their article (what does this mean?). If published, this will include your full peer review and any attached files.

Reviewer #1: No

---

## [Author Response · Author response to Decision Letter 1]

3 Aug 2022

1. The discussion section should include a paragraph on the strengths and weaknesses/limitation of this study. It is alluded to in the last sentence of the conclusion, although, I feel this is placed in the wrong section.

Response: 

Correction has been done. New paragraph has been added and marked in blue, pg. 15, line 290-296

---

## [Decision Letter · Decision Letter 2]

22 Aug 2022

PONE-D-21-08579R2Counselling experiences among men sex men living with HIV in MalaysiaPLOS ONE

Dear Dr. Ruhani,

Thank you for submitting your manuscript to PLOS ONE. After careful consideration, we feel that it has merit but does not fully meet PLOS ONE’s publication criteria as it currently stands. Therefore, we invite you to submit a revised version of the manuscript that addresses the points raised during the review process.

We look forward to receiving your revised manuscript.

Kind regards,

Nelsensius Klau Fauk, S.Fil., M., MHID, MSc, PhD

Academic Editor

PLOS ONE

Journal Requirements:

Additional Editor Comments:

The authors have addressed the comments of the reviewers. I have only a few small things for them to add and then the manuscript can be officially accepted for publication.

I agree with the reviewer, the authors need to carefully read the manuscript and improve the language, including typos, etc.

Lines 35-36: “Factors contributing to the vulnerability of MSM living with HIV are related to stigma and hidden identity”

Use the following reference to support it:

**Culture, social networks and HIV vulnerability among men who have sex with men in Indonesia. PLoS ONE. 2017;12(6):1-14.**

Lines 46-48: “MSM are widely exposed to behavioural risks of HIV infection because they have multiple sexual partners and have been found to consistently engage in casual sex without using condoms”.

Use the following reference to support it:

**“Exploring determinants of unprotected sexual behaviours favouring HIV transmission among men who have sex with men in Yogyakarta, Indonesia. Global Journal of Health Science. 2017;9(8):47-56”.**

Reviewers' comments:

Reviewer's Responses to Questions

**Comments to the Author**

1. If the authors have adequately addressed your comments raised in a previous round of review and you feel that this manuscript is now acceptable for publication, you may indicate that here to bypass the “Comments to the Author” section, enter your conflict of interest statement in the “Confidential to Editor” section, and submit your "Accept" recommendation.

Reviewer #1: All comments have been addressed

2. Is the manuscript technically sound, and do the data support the conclusions?

Reviewer #1: Yes

3. Has the statistical analysis been performed appropriately and rigorously? 

Reviewer #1: N/A

4. Have the authors made all data underlying the findings in their manuscript fully available?

Reviewer #1: Yes

5. Is the manuscript presented in an intelligible fashion and written in standard English?

Reviewer #1: Yes

6. Review Comments to the Author

Reviewer #1: While a paragraph on strengths and limitations has been provided, this paragraph may require copy editing before publication is considered

7. PLOS authors have the option to publish the peer review history of their article (what does this mean?). If published, this will include your full peer review and any attached files.

Reviewer #1: No

---

## [Author Response · Author response to Decision Letter 2]

24 Aug 2022

1. Lines 35-36: “Factors contributing to the vulnerability of MSM living with HIV are related to stigma and hidden identity”

Use the following reference to support it:

Culture, social networks and HIV vulnerability among men who have sex with men in Indonesia. PLoS ONE. 2017;12(6):1-14.

Lines 46-48: “MSM are widely exposed to behavioural risks of HIV infection because they have multiple sexual partners and have been found to consistently engage in casual sex without using condoms”.

Use the following reference to support it:

“Exploring determinants of unprotected sexual behaviours favouring HIV transmission among men who have sex with men in Yogyakarta, Indonesia. Global Journal of Health Science. 2017;9(8):47-56”.

Response

New references have been added, line 37 and line 50. 

New references have been added at the refences section, reference no.3 and no.9.

2. Reviewer #1: While a paragraph on strengths and limitations has been provided, this paragraph may require copy editing before publication is considered

Response

This section has been proof edited and the whole manuscript has been proof edited on 23/8/2022

---

## [Editor Report · Decision Letter 3]

25 Aug 2022

Counselling experiences among men having sex with men and living with HIV in Malaysia

PONE-D-21-08579R3

Dear Dr. Ruhani,

We’re pleased to inform you that your manuscript has been judged scientifically suitable for publication and will be formally accepted for publication once it meets all outstanding technical requirements.

Kind regards,

Nelsensius Klau Fauk, S.Fil., M., MHID, MSc, PhD

Academic Editor

PLOS ONE

---

## [Editor Report · Acceptance letter]

31 Aug 2022

PONE-D-21-08579R3 

Counselling experiences among men having sex with men and living with HIV in Malaysia 

Dear Dr. Mat Min:

I'm pleased to inform you that your manuscript has been deemed suitable for publication in PLOS ONE. Congratulations! Your manuscript is now with our production department. 

Kind regards, 

on behalf of

Dr. Nelsensius Klau Fauk 

Academic Editor

PLOS ONE